# A Diagonal Structured State Space Model on Loihi 2 for Efficient Streaming Sequence Processing

Svea Marie Meyer[1,2], Philipp Weidel[1], Philipp Plank[1], Leobardo Campos-Macias[1],
Sumit Bam Shrestha[1], Philipp Stratmann[1], Jonathan Timcheck[1], and Mathis Richter[1]

[1]Intel Labs, Intel Deutschland GmbH, 85579 Neubiberg, Germany
[2]Institute of Informatics, LMU Munich, 80538 Munich, Germany
`svea.meyer@campus.lmu.de`

## Abstract

Deep State Space Models (SSM) demonstrate state-of-the-art performance on long-range sequence modeling tasks. While the recurrent structure of SSMs can be efficiently implemented as a convolution or as a parallel scan during training, recurrent token-by-token processing cannot currently be implemented efficiently on GPUs. Here, we demonstrate efficient token-by-token inference of the SSM S4D on Intel's Loihi 2 state-of-the-art neuromorphic processor. We compare this first-ever neuromorphic-hardware implementation of an SSM on sMNIST, psMNIST, and sCIFAR to a recurrent and a convolutional implementation of S4D on Jetson Orin Nano (Jetson). While we find Jetson to perform better in an offline sample-by-sample based batched processing mode, Loihi 2 outperforms during token-by-token based processing, where it consumes 1000 times less energy with a 75 times lower latency and a 75 times higher throughput compared to the recurrent implementation of S4D on Jetson. This opens up new avenues towards efficient real-time streaming applications of SSMs.

## 1 Introduction

The attention mechanism has been pervasive in enabling the incredible AI capabilities of today, such as intelligent chatbots [1], object detection [2], and multimodal video understanding [3]. While it excels at associating context at different points in time and building a higher level understanding, the computational cost of the attention mechanism increases quadratically with the context length [4], making it infeasible for very long sequences. Different techniques realize more efficient computation of attention [5, 6, 7] but the fundamental limitation of increased cost for higher context length remains.

Recurrent networks, on the other hand, increase the compute cost only linearly with context length; they are, however, difficult to train. Recently, a family of linear recurrent architectures, SSMs, based on the memory property of state-space dynamics [8, 9, 10, 11, 12] have emerged as an alternative to the attention mechanism. With their recurrent formulation, they offer linearly increasing computational cost, while also being easily trainable using the convolution view of the same model [8]. They have shown superior performance compared to attention-based models on very long-range context [9, 12, 13], while also maintaining competitive performance on language modeling tasks [12].

The recurrent formulation of SSMs with their local stateful computation aligns well with the architecture of neuromorphic processors, in which compute and memory are co-located [14]. This is in contrast to GPUs, where the separation of compute and memory makes them efficient only when processing highly structured compute and memory reads [15] (e.g., batching, convolution) and therefore inefficient when computing the recurrent formulation of SSMs. Previous simulated

Second Workshop on Machine Learning with New Compute Paradigms at NeurIPS 2024(MLNCP 2024).

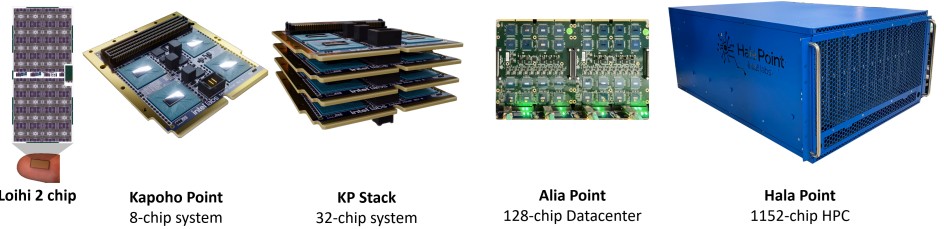



**Loihi 2 chip**     **Kapoho Point**     **KP Stack**     **Alia Point**     **Hala Point**
8-chip system     32-chip system     128-chip Datacenter     1152-chip HPC



Figure 1: Loihi 2 systems of different form factors.

implementations of neuromorphic SSMs have demonstrated a reduction in the number of operations but focus on biologically plausible spiking neurons rather than leveraging performance-optimized neuron models available in modern neuromorphic processors [16, 17, 18].

In this paper, we demonstrate the first implementation of an SSM, S4D, on a neuromorphic processor, Intel's Loihi 2. We compare its accuracy on sequential processing datasets to state-of-the-art methods and benchmark its computational costs against implementations on an edge GPU.

## 2 Background

### 2.1 Deep State Space Models

Deep SSMs have been increasingly used to overcome the challenges of transformers in modeling long sequences [19], starting with the structured SSM, S4, developed by Gu et al. [8]. S4 models can perform their computations using one of three representations, which can be transformed into each other and serve different functional purposes [8, 12]:

$$\dot{\boldsymbol{x}}(t) = \boldsymbol{A}\boldsymbol{x}(t) + \boldsymbol{B}u(t), \qquad\qquad y(t) = \boldsymbol{C}\boldsymbol{x}(t) \tag{1}$$

$$\boldsymbol{x}_k = \overline{\boldsymbol{A}}\boldsymbol{x}_{k-1} + \overline{\boldsymbol{B}}u_k, \qquad\qquad y_k = \overline{\boldsymbol{C}}\boldsymbol{x}_k \tag{2}$$

$$\overline{\boldsymbol{K}} = (\overline{\boldsymbol{C}\boldsymbol{B}}, \overline{\boldsymbol{C}\boldsymbol{A}\boldsymbol{B}}, \cdots, \overline{\boldsymbol{C}\boldsymbol{A}}^{L-1}\overline{\boldsymbol{B}}), \qquad (\cdots, y_k, \cdots, y_L) = \overline{\boldsymbol{K}} * (\cdots, u_k, \cdots, x_L) \tag{3}$$

The *continuous recurrent* representation in Equation (1) processes continuous 1-D signals $u(t)$ to output signals $y(t)$ via an $N$-dimensional latent space $\boldsymbol{x}(t)$: $u(t) \in \mathbb{R} \to \boldsymbol{x}(t) \in \mathbb{R}^N \to y(t) \in \mathbb{R}$. The *discrete recurrent* representation in Equation (2) assumes constant step sizes $\Delta$ to transform the matrices $\boldsymbol{A}$, $\boldsymbol{B}$, and $\boldsymbol{C}$ into discrete matrices $\overline{\boldsymbol{A}}$, $\overline{\boldsymbol{B}}$, and $\overline{\boldsymbol{C}}$ and enables fast autoregressive inference when inputs $u_k$ are presented sequentially. The *convolutional* representation in Equation (3) transforms the linear time-invariant SSM in Equation (1) into a global convolution, which enables efficient, parallelized training when $L$ data points are available in a batch.

Long-range sequence modeling has seen algorithmic advancements, including Liquid S4 [10], S5 [9], and the S6 layers in Mamba [12]. These models have achieved state-of-the-art accuracy in sequence modeling on tasks including speech commands [10], the Path-X version of the Pathfinder challenge [9], autoregressive language modeling, audio waveforms, and DNA sequences [12].

Several hardware-aware adjustments have been applied to SSMs to make them more efficient on GPUs. Most notably, it has been shown that $\boldsymbol{A}$ can be diagonalized with little to no detrimental effect on the algorithmic performance [20], leading to the S4 variant *S4D* [11] used in the present paper.

### 2.2 Intel's Neuromorphic Processor Loihi 2

Intel's Loihi 2 neuromorphic processor (Figure 1) is a fully asynchronous, digital, scalable, event-driven architecture, designed for efficient sparse signal processing. Each Loihi 2 chip features an interconnect of compute and memory co-located computational units, neurocores, that communicate using spiking events and support up to 8192 independent, programmable neurons. The neurons can be stateful, enabling an efficient implementation of self-recurrent dynamics, such as those of S4D. Neurons can output *graded spike events* that carry an integer value. Synaptic connectivity between

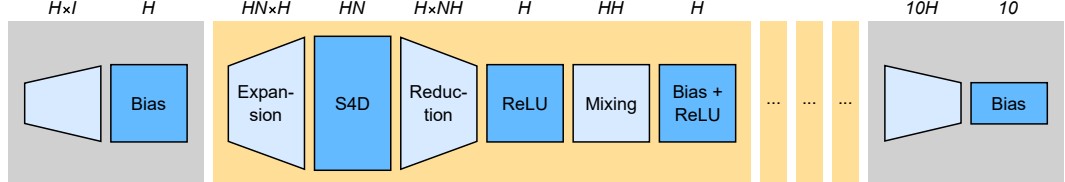

Figure 2: S4D model architecture as implemented on Loihi 2. Light blue layers refer to connections and dark blue layers to programmable neurons on Loihi 2. Variables above each layer denote the dimensionality of the layer.

neurons supports memory-efficient representations of sparse weight matrices to take advantage of unstructured connectivity sparsity.

While the cores and chips of Loihi 2 system execute their computations asynchronously, they synchronize via messages to adhere to a global, algorithmic time-step. At each time-step, the output of a layer of neurons is sent only to its neighboring layer, not through the entire network. For feed-forward networks, input can be injected into the network *pipelined*, at every time-step, for maximum throughput. To avoid generating too much traffic in the network, input can be injected less often; minimum latency is often achieved by *fall-through* processing, where input is injected only once the previous input has traveled through the entire network. The distinction between pipelined and fall-through execution becomes most significant for networks with multiple layers of processing.

The merit of the Loihi 2 architecture has been demonstrated in a variety of domains, including optimization [21, 22], depth estimation [23], and efficient video and audio processing [24].

## 3 Neuromorphic Diagonal Deep State-Space Model

### 3.1 Model architecture on Loihi 2

Figure 2 shows the model architecture of S4D as implemented on Loihi 2. It consists of an encoder layer that expands the input to a higher dimensionality, four S4D blocks, and a decoder layer that reduces the dimensionality to the number of classes. At the top of the figure, the dimensionality of each layer of the model is listed, where $I$ represents the input dimensionality, $H$ is the model dimension, and $N$ is the number of hidden states per model dimension; the output dimensionality is 10. We evaluate two model sizes with 67k parameters ($H = 64, N = 32$) and 265k parameters ($H = 128, N = 64$) for different datasets (see Section 4 for details).

Each S4D block consists of a simplified variant of the S4D model, computing the SSM dynamics as a recurrent neural network (Equation (2)). In contrast to the original S4D model, we only use ReLU activations instead of GLUs and GeLUs to increase activation sparsity. To further simplify the model, we also leave out normalization layers and residual connections. After each S4D layer, the dimensions are mixed using a linear projection followed by another ReLU activation.

All S4D layers, ReLU activations, and biases (depicted in dark blue in Figure 2), are implemented as programmable neurons on Loihi 2. As the matrices $\overline{A}, \overline{B}$, and $\overline{C}$ of the S4D dynamics are fully diagonal, all hidden states in the S4D layers compute their dynamics fully independently without interactions with other states. Therefore, the state-space dynamics can be computed within the programmable neurons, and without using self-recurrent connections or connections between them.

All projections (depicted in light blue in Figure 2), up-projection, expansion, reduction, mixing, and down-projection, are implemented in Loihi 2 using linear synapses in the neurocore alongside their respective neuron instances.

We optimize how each layer is distributed across neurocores to achieve uniform compute load. We place subsequent layers onto neighboring neurocores to reduce message passing time, leading to a usage of 31 and 111 neurocores of a single Loihi 2 chip for the small and large model, respectively.

Table 1: Comparison against leading reported test accuracies from prior works (Transformer, CNN, RNN, SSM) on the sMNIST, psMNIST, and sCIFAR datasets.

| Model (Input length) | sMNIST (784) | psMNIST (784) | sCIFAR (1024) |
|---|---|---|---|
| Transformer [4, 26] | 98.9 | 97.9 | 62.2 |
| CCNN [27] | **99.72** | **98.84** | **93.08** |
| LipschitzRNN [28] | 99.4 | 96.3 | 64.2 |
| LSSL [29] | 99.53 | 98.76 | 84.65 |
| S4 [8, 11] | 99.63 | 98.70 | 91.80 |
| S4D [11] | - | - | 90.69 |
| Liquid-S4 [10] | - | - | 92.02 |
| S5 [9] | 99.65 | 98.67 | 90.10 |
| AHP SNN on Loihi 1 [9] | 96.00 | - | - |
| S4D, full precision (**Ours**) | 99.51 | 97.53 | 86.53 |
| S4D, after PTQ (**Ours**) | 99.20 | 92.45 | 71.74 |
| **S4D, on Loihi 2 after QAFT (Ours)** | 99.20 | 96.16 | 84.13 |

## 3.2 Post Training Quantization and Quantization Aware Fine Tuning

The S4D models are first trained in full precision in convolution mode. Since Loihi 2 only supports fixed precision computation, fake quantization is applied after the initial training. In this post-training quantization (PTQ) step, we use 8-bit weights, 24-bit spike messages, and 24-bit neuron states (including the $\overline{A}$, $\overline{B}$, and $\overline{C}$ representation). Subsequently, we perform quantization aware fine tuning (QAFT) in recurrent mode to recover any loss in accuracy due to PTQ.

In the fake quantization scheme with $b$-bit representation, we quantize the tensor $X$ as $\hat{X} = \lfloor X s \rceil \frac{1}{s}$ with the scaling factor $s = 2^{b-1}/X_{\max}$. The bound $X_{\max}$ is selected analytically for state space parameters like $\overline{A}$, while for weights and biases it is computed as $|X|_{\max}$ to maximize the dynamic representation of the integer representation. To be fully accurate to Loihi 2's fixed precision arithmetic, we also quantize the descaling factor $1/s$ with a precision of 16-bit. To allow gradient flow in the backward computation, we use a straight-through estimator [25].

## 3.3 Reference evaluation on Jetson Orin Nano

As a reference, we implement both the convolution and the recurrent formulation of our S4D variant on Jetson. Due to lack of complex tensor support in TensorRT, the evaluations are performed using a PyTorch just-in-time compiled model using fp32 precision. The power measurement reported by the jtop API is used for characterization. For runtime, only the time spent to input the data and compute the output was considered. The primary point of comparison for Loihi 2 vs Jetson implementation is the streaming (batch=1) mode of inference. In addition to the streaming mode of inference, the peak-performing batched mode of inference is included for a representative optimum performance on Jetson.

# 4 Results

We evaluate our S4D model running on Intel Loihi 2 and Nvidia's Jetson on the datasets sequential MNIST (sMNIST), permuted sequential MNIST (psMNIST), and sequential CIFAR10 (sCIFAR).

## 4.1 Accuracy and parameter count

Table 1 shows the accuracy on sMNIST, psMNIST, and sCIFAR of our S4D model in full precision, after quantization, and on Loihi 2 in comparison to other models.

Although we use a simplified version of the S4D model, by only using ReLU activations and no normalization (see Section 3.1), the performance in full precision meets that of the original and

Table 2: Performance comparison.

| | HW | Exec mode | Prec | Acc (↑) (%) | Token-by-Token Processing | | | | Sample-by-Sample Processing | | | |
|---|---|---|---|---|---|---|---|---|---|---|---|---|
| | | | | | Energy (↓) (mJ/token) | Latency (↓) (ms/token) | Throughput (↑) (token/s) | EDP (↓) (μJ s/token) | Energy (↓) (mJ/sample) | Latency (↓) (ms/sample) | Throughput (↑) (sample/s) | EDP (↓) (μJ s/sample) |
| sMNIST | Loihi 2* | Rec (ft) | qint | 99.20 | 0.003 | **0.068** | 14,705 | **0.0002** | 2.678 | 53.314 | 19 | 141.59 |
| | Loihi 2* | Rec (pipe) | qint | 99.20 | **0.002** | 0.168 | **83,343** | 0.0004 | 1.828 | 9.575 | 106 | 17.502 |
| | Jetson† | Rec (b=1) | fp32 | **99.51** | 15.725 | 4.976 | 201 | 79.252 | 12328.652 | 3901.313 | 0.256 | $48.097 \times 10^6$ |
| | Jetson† | Conv (b=1) | fp32 | **99.51** | 23.000 | 6.366 | 157 | 146.418 | 23.000 | **6.366** | 157 | 146.418 |
| | Jetson† | Conv (b=256) | fp32 | **99.51** | - | - | - | - | **0.217** | 8.872 | **28,853** | **1.921** |
| psMNIST | Loihi 2* | Rec (ft) | qint | 96.16 | 0.003 | **0.068** | 14,720 | **0.0002** | 2.678 | 53.262 | 19 | 142.639 |
| | Loihi 2* | Rec (pipe) | qint | 96.16 | **0.002** | 0.168 | **83,349** | 0.0004 | 1.920 | 9.574 | 106 | 15.200 |
| | Jetson† | Rec (b=1) | fp32 | **97.53** | 15.851 | 5.012 | 200 | 70.449 | 12426.807 | 3929.739 | 0.254 | $48.834 \times 10^6$ |
| | Jetson† | Conv (b=1) | fp32 | **97.53** | 23.183 | 6.306 | 158 | 146.187 | 23.183 | **6.306** | 158 | 146.187 |
| | Jetson† | Conv (b=256) | fp32 | **97.53** | - | - | - | - | **0.218** | 8.837 | **28,969** | **1.924** |
| sCIFAR | Loihi 2* | Rec (ft) | qint | 84.13 | 0.016 | **0.066** | 15,259 | **0.0010** | 16.284 | 65.534 | 15 | 1092.808 |
| | Loihi 2* | Rec (pipe) | qint | 84.13 | **0.010** | 0.172 | **81,508** | 0.0017 | 10.355 | 12.735 | 80 | 131.869 |
| | Jetson† | Rec (b=1) | fp32 | **86.53** | 16.106 | 4.978 | 201 | 80.173 | 16492.163 | 5097.42 | 0.194 | $84.067 \times 10^6$ |
| | Jetson† | Conv (b=1) | fp32 | **86.53** | 26.887 | 6.325 | 158 | 170.053 | 26.887 | **6.325** | 158 | 170.053 |
| | Jetson† | Conv (b=64) | fp32 | **86.53** | - | - | - | - | **0.961** | 8.476 | **7,550** | **8.142** |

\* Loihi 2 workloads were characterized on an Oheo Gulch system with N3C2-revision Loihi 2 chips running on NxCore 2.5.8 and alpha version of the NxKernel API with on-chip IO unthrottled sequencing of input tokens.

† GPU workloads were characterized on an NVIDIA Jetson Orin Nano 8GB 15W TDP running Jetpack 5.1.2, TensorRT 8.6.1, Torch-TensorRT 1.3.0. Energy values include CPU_GPU_CV and SOC components as reported by jtop.

‡ Performance results are based on testing as of September 2024 and may not reflect all publicly available security updates. Results may vary.

more complex S4 and S4D model; only on the sCIFAR dataset, the accuracy of our model is lower with 86.53 %, compared to the original 90.69 %. When we quantize the model after training (PTQ), the accuracy only drops substantially on the psMNIST (97.53 % to 92.45 %) and sCIFAR (86.53 % to 71.74 %) datasets. This loss in accuracy can be recovered to nearly the level of the full precision model by applying quantization-aware fine tuning (QAFT) (psMNIST: 96.16 %, sCIFAR: 84.13 %).

Previous neuromorphic solutions such as the AHP-based model on Loihi 1 [30] reach a substantially lower accuracy, demonstrating the maturing of models and hardware in the neuromorphic domain. Overall, the CCNN model exhibits the highest accuracy on all tasks, while using 2M parameters [27]. Our model shows competitive performance with less than 265k parameters for sCIFAR and 67k parameters for the sMNIST datasets.

## 4.2 Computational cost

Table 2 reports computational cost of inference on Loihi 2 and Jetson on the three datasets. In addition to energy, latency, and throughput, the energy-delay product (EDP) [24] is reported to compare systems running at different speed. For these measurements on Loihi 2, we store a sequence of input values on a neurocore and inject them to the S4D network at peak capacity without IO constraints to obtain stable power measurements; this is repeated for 10 representative samples. The energy of the neurocore used to store input values is included in the results for Loihi 2, while the IO power of Jetson is excluded.

**Sample-by-sample processing** The right side of Table 2 shows results from sample-by-sample processing, which assumes that all tokens of a sample are available to the system at the beginning of processing and only a single classification is required for the whole sample. Recurrent formulations of the model have to process each token sequentially, while convolutional formulations can process the entire sample with a single convolution.

In the online processing regime of batch=1, it is evident that Loihi 2 is very efficient compared to the recurrent mode on Jetson and shows better energy per sample of 1.8 mJ compared to 23 mJ for Jetson in the convolutional mode that is favorable for GPU architectures. The throughput and latency between Loihi 2 and Jetson in convolutional mode are also competitive. Jetson, however, achieves peak performance in higher batch mode with substantially reduced energy per sample of

0.22 mJ and 0.96 mJ for sMNIST and sCIFAR along with orders of magnitude higher throughput, which is expected for GPU architectures.

**Token-by-token processing**   The middle columns of Table 2 show results of token-by-token processing. This assumes streaming input that requires a classification for every token. For these types of tasks, both Loihi 2 and the recurrent mode on Jetson process all tokens sequentially. The convolution mode on Jetson, however, has to perform a convolution over the entire sequence with every new token. Its latency, energy, throughput, and EDP are therefore the same for processing one token in token-by-token processing as they are for processing one sample in the sample-by-sample processing mode when using the convolutional implementation.

For token-by-token processing, Loihi 2 outperforms Jetson in all metrics on all datasets. Latency and energy are two to three orders of magnitude lower for Loihi 2. This is reflected in EDP: For MNIST workloads, EDP for Loihi 2 is 0.0002 µJ s and 0.001 µJ s for sCIFAR. In contrast, the recurrent processing on Jetson incurs EDP of 70 µJ s and 80 µJ s on the respective datasets, demonstrating the efficiency of Loihi 2 in token-by-token processing.

**Fall-through vs. pipelined processing**   Table 2 shows the tradeoff between latency and throughput when executing the model on Loihi 2 in fall-through or pipelined processing (see Section 2.2). With fall-through processing, we see a latency of 68 µs per token on the sMNIST dataset, compared to 168 µs in pipelined processing. This comes at the cost of throughput of only 14.705 token/s, compared to 83.343 token/s in pipelined processing. The lower throughput per token in fall-through mode results in a higher latency per sample of 53.314 ms compared to the pipelined mode with 9.57 ms. Results on the other datasets highlight the same tradeoff.

## 5   Discussion

The results in Section 4 show that the performance of S4D on Loihi 2 in comparison to edge GPU systems such as the Jetson depends on the specific use-case.

The S4D model on Loihi 2 excels particularly in use-cases in which an incoming data stream must be processed on a token-by-token basis as quickly as possible. In those use-cases, Loihi 2 can leverage its compute and memory co-located architecture to outperform Jetson. On sCIFAR, the largest workload we measured, Loihi 2 consumes 1000 times less energy with a 75 times lower latency and a 75 times higher throughput compared to the recurrent implementation of S4D on Jetson.

Running S4D on Loihi 2 is not competitive with GPU systems for use-cases that involve the offline processing of large amounts of data in parallel. With offline processing, the data is readily available and multiple batches can be processed in parallel. In these cases, the Jetson can fully realize its potential by executing S4D in convolutional mode and outperform Loihi 2 in throughput, energy, and latency.

It should be noted that our implementation of S4D on Jetson is not completely optimized for speed and efficiency due to the lack of support in TensorRT for complex numbers (see Section 3.3). This will be addressed in future work. We do not expect it to change the qualitative findings presented in this paper.

The accuracy of S4D on Loihi 2 could be improved further by applying QAFT for more than one epoch. Direct extensions of S4 (e.g., Liquid-S4 [10], S5 [9]) have shown state-of-the-art performance on sequence modeling tasks; they are compatible with Loihi 2 and could be evaluated in future work.

The balance of latency, energy, and throughput can be further optimized on Loihi 2 for specific use cases. There is a continuum between fall-through and pipelined processing, not explored here. It enables balancing latency and throughput by introducing just enough pipelining such that the throughput matches the sampling rate of the input sequence.

Ultimately, this work and possible optimizations should now be applied and tested in real-world streaming use-cases, such as keyword-spotting, audio denoising, drone control, and other latency or energy constrained domains.

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
