# OpenReview forum: "A Diagonal State Space Model on Loihi 2 for Efficient Streaming Sequence Processing"
_NeurIPS.cc/2024/Workshop/MLNCP — MLNCP Oral_

### Official Review · Reviewer_bKHp · 2024-10-03
**Proposes a chip specific architecture with thorough analysis**

**Rating:** 9
**Confidence:** 3

**Review:**

The paper presents a novel implementation of the Diagonal State Space Model (S4D) on Intel’s Loihi 2 neuromorphic processor, demonstrating its efficiency for token-by-token sequence modeling tasks.

The authors provide a comprehensive performance comparison against both recurrent and convolutional implementations on GPU, covering key metrics like energy, latency, and throughput across various datasets. The energy savings achieved by Loihi 2, particularly in token-by-token processing (1000x less energy consumption), are impressive and validate the potential of neuromorphic processors in streaming and real-time tasks.

Given the performance limitations on Jetson due to TensorRT’s lack of support for complex tensors, what improvements would you anticipate if a fully optimized Jetson implementation were available?

Are there any quantization alternatives to post-training fake quantization and how do they affect the performance.

---

### Official Review · Reviewer_QP2R · 2024-10-04
**First implementation of SSMs on neuromorphic hardware (Loihi 2) describing the full pipeline of the deployment including quantization. Convincing energy and latency results for streaming mode in time series classification.**

**Rating:** 8
**Confidence:** 5

**Review:**

The work shows the first published implementation of state-space models on neuromorphic hardware, here on Loihi 2. The authors implement a modified version of S4D (modified to fit the capabilities of Loihi 2) and evaluate it for 3 time series classification tasks (sMNIST, psMNIST and sCIFAR-10). While the model performance (accuracy) is close to the state-of-the-art model, the implementation is orders of magnitudes more energy-efficient and faster than a reference implementation on a Jetson Orin Nano GPU. The energy and latency measurements are performed for a wide range of scenarios. The results are discussed fairly, highlighting both the scenarios where Loihi 2 performs best and where the Jetson system performs better.

# quality
The research presented in this paper is of high quality showing highly-efficient realizations of SSMs on neuromorphic hardware.

# clarity
The paper is well written and very comprehensible.

Suggested improvement:
- make clear which model size (small or large) is used for which benchmark in tables 1 and 2.

# originality
It could be expected that SSMs can be implemented on a neuromorphic platform like Loihi 2. The authors do not show completely new concepts but they clearly describe the methodology and pipeline of the implementation of the S4D on Loihi 2. They also provide detailed energy and latency evaluations under different scenarios in comparison with a Jetson Nano.

# significance
The work highlights the suitability of neuromorphic hardware in general, and Loihi 2 in particular, to implement SSMs for energy-efficient and low-latency time series classification in streaming situations, outperforming the solution on Jetson Orin Nano by orders of magnitude.

# pros:
- hardware implementation of SSM on neuromorphic hardware
- includes energy and latency results
- includes comparison to Jetson Orin Nano
- fair study and discussion showing also the cases where the GPU performs better

# cons
- none

---

### Decision · Program_Chairs · 2024-10-10

Accept (Oral)